# Scalable Message Passing Neural Networks: No Need for Attention in Large Graph Representation Learning

**Haitz Sáez de Ocáriz Borde**
University of Oxford

**Artem Lukoianov**
Massachusetts Institute of Technology

**Anastasis Kratsios**
McMaster University & the Vector Institute

**Michael Bronstein**
University of Oxford & AITHYRA

**Xiaowen Dong**
University of Oxford

## Abstract

We propose Scalable Message Passing Neural Networks (SMPNNs) and demonstrate that, by integrating standard convolutional message passing into a Pre-Layer Normalization Transformer-style block instead of attention, we can produce high-performing deep message-passing-based Graph Neural Networks (GNNs). This modification yields results competitive with the state-of-the-art in large graph transductive learning, particularly outperforming the best Graph Transformers in the literature, without requiring the otherwise computationally and memory-expensive attention mechanism. Our architecture not only scales to large graphs but also makes it possible to construct deep message-passing networks, unlike simple GNNs, which have traditionally been constrained to shallow architectures due to oversmoothing. Moreover, we provide a new theoretical analysis of over-smoothing based on universal approximation which we use to motivate SMPNNs. We show that in the context of graph convolutions, residual connections are necessary for maintaining the universal approximation properties of downstream learners and that removing them can lead to a loss of universality.

## 1 Introduction

Traditionally, Graph Neural Networks (GNNs) (Scarselli et al., 2009) have primarily been applied to model functions over graphs with a relatively modest number of nodes. However, recently there has been a growing interest in exploring the application of GNNs to large-scale graph benchmarks, including datasets with up to a hundred million nodes (Hu et al., 2020). This exploration could potentially lead to better models for industrial applications such as large-scale network analysis in social media, where there are typically millions of users, or in biology, where proteins and other macromolecules are composed of a large number of atoms. This presents a significant challenge in designing GNNs that are scalable while retaining their effectiveness.

To this end, we take inspiration from the literature on Large Language Models (LLMs) and propose a simple modification to how GNN architectures are typically arranged. Our framework, Scalable Message Passing Neural Networks (SMPNNs), enables the construction of deep and scalable architectures that outperform the current state-of-the-art models for large graph benchmarks in transductive classification. More specifically, we find that following the typical construction of the Pre-Layer Normalization (Pre-LN) Transformer formulation (Xiong et al., 2020) and replacing attention with standard message-passing convolution is enough to outperform the best Graph Transformers in the literature. Moreover, since our formulation does not necessarily require attention, our architecture scales better than Graph Transformers. Attention can also be easily incorporated into our framework if needed; however, in general, we find that adding attention, at least for large-scale graph

transductive learning, only leads to marginal improvements in performance at the cost of being more computationally demanding.

Our empirical observations, which demonstrate that SMPNNs can use many layers unlike traditional GNNs, are supported by recent theoretical studies on oversmoothing and oversharpening in graph convolutions (Giovanni et al., 2023) and Transformers (Dovonon et al., 2024). These studies suggest the crucial role of residual connections in mitigating oversmoothing and low-frequency dominance in representations. It is worth noting, however, that the aforementioned works primarily approached this issue from a theoretical standpoint and did not scale to large graph datasets. Expanding upon previous theoretical studies on oversmoothing, we provide a universal approximation perspective. Specifically, we demonstrate that residual connections, such as those found in the Transformer block architecture and other newer blocks utilized in the LLM literature, like the Mamba block (Gu & Dao, 2023), are essential for preserving the universal approximation properties of downstream classifiers.

**Contributions.** We propose SMPNNs, a framework designed to scale traditional message-passing GNNs. Our main contributions are the following: (i) The SMPNN architecture can *scale to large graphs* thanks to its $\mathcal{O}(E)$ graph convolution computational complexity, outperforming state-of-the-art Graph Transformers for transductive learning without using global attention mechanisms. It also enables *deep message-passing* GNNs without suffering from oversmoothing, a problem that has traditionally limited these networks to shallow configurations. (ii) We theoretically analyze the advantages of our architecture compared to traditional convolutions over graphs from a universal approximation perspective. Importantly, unlike previous works, we do not rely on the asymptotic convergence of the system to explain its behavior. (iii) We perform extensive experiments in large-graph transductive learning as well as on smaller datasets and demonstrate that our model consistently outperforms recently proposed Graph Transformers and other traditional scalable architectures. We also conduct ablations and additional experiments to test the importance of the different components of our architecture.

## 2   BACKGROUND

**Graph Convolutional Networks (GCNs)** implement a special type of message passing of the following form: $\mathbf{X}^{(l+1)} = \zeta\left(\tilde{\mathbf{A}}\mathbf{X}^{(l)}\mathbf{W}^{(l)}\right)$, where $\zeta$ is an activation function, $\mathbf{W}^{(l)}$ is a learnable weight matrix, and $\tilde{\mathbf{A}}$ is computed as a function of the degree matrix and the adjacency matrix. The features are aggregated row-wise: $\left(\tilde{\mathbf{A}}\mathbf{X}^{(l)}\right)_i = c_{ii}\mathbf{x}_i^{(l)} + \sum_{j\in\mathcal{N}(v_i)} c_{ij}\mathbf{x}_j^{(l)}$; $c_{ij} = \frac{1}{\sqrt{\deg(v_i)\deg(v_j)}}$.

**Transformers.** The Transformer architecture (Vaswani et al., 2017) is characterized by its all-to-all pairwise communication between nodes via the scaled dot-product attention mechanism. Transformers can be seen as an MPNN on a fully-connected graph (Sáez de Ocáriz Borde, 2024; Bronstein et al., 2021), where all nodes are within the one-hop neighborhood of each other in every layer. This avoids issues in capturing long-range dependencies inherent to message passing, since they are all connected. However, Transformers discard the locality inductive bias of the input graph. Given input query, key, and value matrices, $\mathbf{Q} = \mathbf{X}\mathbf{W}_Q$, $\mathbf{K} = \mathbf{X}\mathbf{W}_K$, $\mathbf{V} = \mathbf{X}\mathbf{W}_V \in \mathbb{R}^{N\times D}$ (where for simplicity we assume the feature dimensionality of these matrices is kept the same as the original node features, $\mathbf{X} \in \mathbb{R}^{N\times D}$), the self-attention operation in the encoder blocks computes: $\mathbf{X}' = \mathrm{softmax}(\mathbf{Q}\mathbf{K}^T/\sqrt{D})\mathbf{V}$. Attention is challenging to scale to large graphs due to its computational complexity of $\mathcal{O}(N^2)$ in the number of nodes. Large-scale Graph Transformers typically replace this operation with linear attention to avoid GPU memory overflow.

**Large Graph Transformers.** The emergence of large-scale graph-structured datasets such as social networks has led to increased recent interest in scaling GNNs to very large graphs with up to hundreds of millions of nodes. Large-scale graph learning tasks are often transductive learning settings, where one is given a fixed graph with partially labeled nodes and attempts to infer the properties of the missing nodes. While Graph Transformers currently produce state-of-the-art results, their main challenge is the computational complexity of attention for a large number of nodes, which is equivalent to very long context windows in LLMs. Techniques aiming at solving this problem include architectures such as Nodeformer (Wu et al., 2022), which utilizes a kernelized Gumbel-Softmax operator, DIFFormer (Wu et al., 2023a), which builds on recent advances in graph diffusion, and SGFormer (Wu et al., 2023b), which utilizes a single linear "attention" operation without the soft-

max activation to avoid computational overhead when scaling to graphs with up to a hundred million nodes. Other architectures such as Exphormer (Shirzad et al., 2023) use sparse attention mechanisms based on expander graphs to aid scaling. Additionally, Exphormer also uses linear global attention with respect to a virtual node in parallel to the expander graph. Some of the aforementioned architectures rely on the GPS Graph Transformer framework (Rampášek et al., 2022), which combines attention with standard message-passing GNN layers and computes both in parallel in the form of a hybrid architecture. Other scalable architectures that are not based on Transformers, such as SGC (Wu et al., 2019) and SIGN (Frasca et al., 2020), have been developed for large-scale graphs, but their performance is generally worse than that of Graph Transformers, as shown in Section 5.

Additional background on graph learning and message passing is provided in Appendix A.

# 3    SCALABLE MESSAGE PASSING NEURAL NETWORKS

In the GNN literature, it has been shown that graph convolutions can exhibit asymptotic behaviors other than over-smoothing in the limit of many layers when equipped with residual connections (Giovanni et al., 2023). This theoretical observation aligns with well-known best practices in the LLM community (Vaswani et al., 2017; Touvron et al., 2023; Brown et al., 2020), which is highly experienced with both large-scale datasets and models. In this work, we aim to bridge the gap between the GNN and LLM literature and benefit from the cross-pollination of ideas.

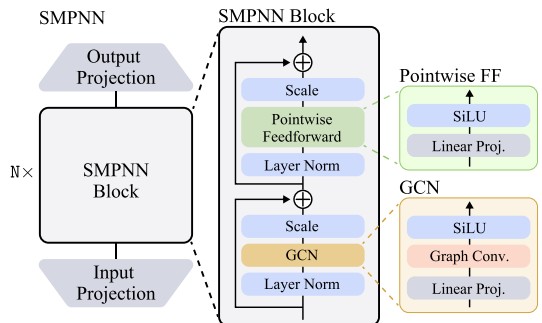

Figure 1: **The Scalable Message Passing Neural Network (SMPNN) architecture.** *Left:* The full model is comprised of N transformer-style blocks stacked one after the other. The model also uses input and output feedforward layers to project node features to the hidden and output dimensions. *Middle:* Architecture of a single SMPNN block as described in Section 3.1. *Right:* Zoom into the GCN block and the pointwise feedforward network with SiLU activation functions.

**Packaging Attention and Message-Passing.** Attention has been much emphasized in the literature and, although it is certainly paramount, the importance of the other components around it should not be underestimated. Interestingly, in retrospect, the ubiquitous attention mechanism used in LLMs had already been proposed (Bahdanau et al., 2016) before the Transformer architecture (Vaswani et al., 2017); however, it was not until it was "packaged" into the Transformer block as we know it today, that it really led to the recent breakthroughs in language modeling. The scaled-dot-product attention mechanism (and its variants) in Transformers can be seen as message-passing attentional GNNs over a complete graph (Sáez de Ocáriz Borde, 2024; Bronstein et al., 2021). It is well-known among practitioners in the LLM community that this operation struggles to learn effectively without residual connections. Yet, traditional GNN architectures simply stack message-passing layers one after the other (Kipf & Welling, 2017; Schlichtkrull et al., 2017; Veličković et al., 2018), which seems to contradict the modus operandi of large-scale model engineering.

*The main motivation of our paper is to demonstrate that the standard architectural packaging approaches used in NLP and LLMs are also applicable to GNNs, and that the standard attention mechanism used in Transformers can be substituted with message-passing layers.*

## 3.1    ARCHITECTURE DESIGN AND THE SCALABLE MESSAGE PASSING BLOCK

Drawing from theoretical observations in the GNN literature and best practices in language modeling, we propose the following architecture, which is similar to the Transformer and comprises two parts: initial nodewise message-passing, analogous to tokenwise communication, and a pointwise feedforward layer to transform the feature vector of each node in isolation. Unlike the Transformer, however, our architecture has linear rather than quadratic scaling.

**The Transformer Block.** The Transformer architecture has proven surprisingly robust, and the only main modification it has undergone since its inception is the change in the location of normalization, which is now applied before attention. The advantages of applying layer normalization (Ba et al., 2016) before, as in the Pre-LN Transformer, have already been studied both theoretically (Xiong et al., 2020) and empirically in the literature (Baevski & Auli, 2019; Child et al., 2019; Wang et al., 2019), and we will also follow this in our implementation. Before the Transformer, residual connections were already a common component in most deep architectures for vision and language tasks since being popularized by the ResNet model (He et al., 2015). In the LLM literature, even when alternatives to attention have been proposed, such as Mamba (Gu & Dao, 2023), operations that perform token-wise communication (equivalent to message-passing between nodes) are always accompanied by *residual connections*, as well as *linear projections*, *normalization*, and sometimes *gatings*.

**Message-Passing.** We integrate a standard GCN layer into the Pre-LN Transformer-style block. In particular, a single block applies the following sequence of transformations to the input matrix of node features $\mathbf{X}^{(l)} \in \mathbb{R}^{N \times D}$, where $N$ is the number of nodes and $D$ is the dimensionality of the feature vectors:

$$\mathbf{H}_1^{(l)} = \text{LayerNorm}(\mathbf{X}^{(l)}). \tag{1}$$

Next, a GCN layer (plus additional components) is used for nodewise local communication instead of the standard global self-attention:

$$\mathbf{H}_2^{(l)} = \alpha_1^{(l)} \, \text{SiLU}\left(\tilde{\mathbf{A}}\mathbf{H}_1^{(l)}\mathbf{W}_1^{(l)}\right) + \mathbf{X}^{(l)}, \tag{2}$$

where $\tilde{\mathbf{A}}$, the degree-normalized adjacency matrix, is $\tilde{\mathbf{A}}_{i,j} = 1/\sqrt{\deg(v_i)\deg(v_j)}$ if $v_i$ and $v_j$ are neighbors and 0 otherwise. $\text{SiLU}(x) = \frac{x}{1+e^{-x}}$ and it is applied elementwise. We also introduce a scaling factor $\alpha_1^{(l)}$ which is initialized at $10^{-6}$ for applying identity-style block initialization (Peebles & Xie, 2023). Importantly, the layer defined in equation 2 contains a residual connection, the importance of which will be theoretically justified in Section 4.

**Pointwise feedforward.** The second part of the block is a pointwise transformation of the feature vectors preceded by another learnable normalization:

$$\mathbf{H}_3^{(l)} = \text{LayerNorm}(\mathbf{H}_2^{(l)}), \tag{3}$$

$$\mathbf{X}^{(l+1)} = \mathbf{H}_4^{(l)} = \alpha_2^{(l)} \text{SiLU}(\mathbf{H}_3^{(l)}\mathbf{W}_2^{(l)}) + \mathbf{H}_2^{(l)}, \tag{4}$$

where similar to before, we introduce the learnable scaling $\alpha_2^{(l)}$, also initialized at $10^{-6}$.

Our *Scalable Message Passing Neural Network block* is thus of the form $\mathbf{X} \mapsto \mathbf{H}_4$. In Figure 1, we display the complete SMPNN architecture at different levels of granularity, which consists of stacking multiple instances of the aforementioned block. In Appendix B, we discuss how to augment SMPNNs with attention, although we find that this only leads to minor performance improvements, see Table 2 in Section 5.

### 3.2 Computational Complexity and Comparison with Graph Transformers

In Table 1, we compare the computational complexity of SMPNNs to that of various Graph Transformers in the literature. Our model's graph convolution layer inherits the computational cost of GCNs, $\mathcal{O}(E)$, assuming a sparse representation of the adjacency matrix (Kipf & Welling, 2017). Graph Transformers such as SGFormer use both graph convolutions and linear attention, resulting in a total cost of $\mathcal{O}(N + E)$ in terms of nodewise communication operations. Note that although we are highlighting the complexity of graph convolution compared to that of linear attention, $\mathcal{O}(N)$, given our architecture also has other components that act pointwise such as linear layers (like all architectures in general) it is more accurate to think of the overall complexity of SMPNNs as being $\mathcal{O}(N + E)$. Unlike other models, we do not apply $\mathcal{O}(N^3)$ pre-processing steps (Dwivedi & Bresson, 2021; Ying et al., 2021; Chen et al., 2022; Hussain et al., 2022; Rampášek et al., 2022), which would be prohibitively expensive for the large graph datasets we are targeting. Additionally,

our model does not require positional encodings, attention, augmented training loss, or edge embeddings to achieve competitive performance. We would like to highlight that the majority of Graph Transformers in Table 1 have been designed for smaller graphs and have only been demonstrated on datasets with thousands of nodes or fewer due to their quadratic complexity. Our main competitors are therefore NodeFormer (Wu et al., 2022), DIFFormer (Wu et al., 2023a), and SGFormer (Wu et al., 2023b), which have managed to scale to millions of nodes using less expressive or simplified versions of attention without quadratic complexity.

Table 1: Comparison of model components used by different Graph Transformers: Positional Encodings (PE), Multihead Attention (MA), Augmented Training Loss (ATL), and Edge Embeddings (EE). We also report whether the model uses All-Pair communication (typically implemented as some variant of attention), the Pre-processing and Training computational complexity, and the largest graph for which the method's performance has been reported. *Random regular expander graph generation is needed, and graphs that fail to be near-Ramanujan are discarded and regenerated.

| Model | Model Components | | | | All-Pair | Pre-processing | Training | Largest Demo |
|---|---|---|---|---|---|---|---|---|
| | PE | MA | ATL | EE | | | | |
| GraphTransformer (Dwivedi & Bresson, 2021) | ✓ | ✓ | ✗ | ✓ | ✓ | $\mathcal{O}(N^3)$ | $\mathcal{O}(N^2)$ | 0.2K |
| Graphormer (Ying et al., 2021) | ✓ | ✓ | ✗ | ✓ | ✓ | $\mathcal{O}(N^3)$ | $\mathcal{O}(N^2)$ | 0.3K |
| GraphTrans (Jain et al., 2021) | ✗ | ✓ | ✗ | ✗ | ✓ | - | $\mathcal{O}(N^2)$ | 0.3K |
| SAT (Chen et al., 2022) | ✓ | ✓ | ✗ | ✗ | ✓ | $\mathcal{O}(N^3)$ | $\mathcal{O}(N^2)$ | 0.2K |
| EGT (Hussain et al., 2022) | ✓ | ✓ | ✓ | ✓ | ✓ | $\mathcal{O}(N^3)$ | $\mathcal{O}(N^2)$ | 0.5K |
| GraphGPS (Rampášek et al., 2022) | ✓ | ✓ | ✗ | ✓ | ✓ | $\mathcal{O}(N^3)$ | $\mathcal{O}(N+E)$ | 1.0K |
| Gophormer (Zhao et al., 2021) | ✓ | ✓ | ✓ | ✗ | ✗ | - | $\mathcal{O}(Nsm^2)$ | 20K |
| Exphormer (Shirzad et al., 2023) | ✓ | ✓ | ✗ | ✓ | ✓ | * | $\mathcal{O}(N+E)$ | 132K |
| NodeFormer (Wu et al., 2022) | ✓ | ✓ | ✓ | ✗ | ✓ | - | $\mathcal{O}(N+E)$ | 2.0M |
| DIFFormer (Wu et al., 2023a) | ✗ | ✓ | ✗ | ✗ | ✓ | - | $\mathcal{O}(N+E)$ | 1.6M |
| SGFormer (Wu et al., 2023b) | ✗ | ✗ | ✗ | ✗ | ✓ | - | $\mathcal{O}(N+E)$ | 100M |
| SMPNN | ✗ | ✗ | ✗ | ✗ | ✗ | - | $\mathcal{O}(N+E)$ | 100M |

## 4 THEORETICAL JUSTIFICATION

In this section, we theoretically justify our architectural design. We review some theoretical findings from the literature that hint at the importance of residual connections to avoid oversmoothing in message-passing convolutions over graphs in Appendix C. Inspired by these results, we further extend the analysis through the lens of universal approximation, which does not rely on asymptotic behavior, unlike other works such as Giovanni et al. (2023). For the proofs, see Appendix D.

We consider a class of models to be expressive if it is a universal approximator, in the local uniform sense of Hornik et al. (1989); Kidger & Lyons (2020); Duan et al. (2023); Yarotsky (2024), meaning that it can approximately implement any continuous function uniformly on compact sets. We will use $\mathcal{C}(\mathbb{R}^{N \times D})$ to denote the set of continuous functions from $\mathbb{R}^{N \times D}$ to $\mathbb{R}$, when both are equipped with the (only) norm topologies thereon.

**Definition 4.1** (Universal Approximator). *A subset of models $\mathcal{F} \subseteq \mathcal{C}(\mathbb{R}^{N \times D})$ is said to be a universal approximator in $\mathcal{C}(\mathbb{R}^{N \times D})$ if: for each continuous target function $f : \mathbb{R}^{N \times D} \to \mathbb{R}$, each uniform approximation error $\varepsilon > 0$, and every non-empty compact set of inputs $K \subset \mathbb{R}^{N \times D}$, there exists some model $\hat{f} \in \mathcal{F}$ such that*

$$\max_{\mathbf{X} \in K} |f(\mathbf{X}) - \hat{f}(\mathbf{X})| < \varepsilon. \tag{5}$$

In what follows, for any activation function $\zeta : \mathbb{R} \to \mathbb{R}$ we use $\mathcal{MLP}^{\zeta}_{N \times D} = \mathcal{MLP}_{N \times D}$ to denote the class of MLPs with activation function $\zeta$, mapping $\mathbb{R}^{N \times D}$ to $\mathbb{R}$. We consider the following regularity condition on $\zeta$.

**Assumption 4.2** (Regular Activation). *The activation function $\zeta : \mathbb{R} \to \mathbb{R}$ is continuous and either: (i) Pinkus (1999): $\zeta$ is non-polynomial, (ii) Kidger & Lyons (2020): $\zeta$ is non-affine and there exists some $t \in \mathbb{R}$ at which $\zeta$ is continuously differentiable and $\zeta'(t) \neq 0$.*

We will consider two classes of models, highlighting the key benefits and drawbacks of adding versus omitting residual connections in graph convolution layers.

**Definition 4.3** (No Residual Connection Model Class). *Consider the class $\mathcal{F}_{\mathcal{G},\mathbf{W}}$ which consists of all maps $\hat{f} : \mathbb{R}^{N \times D} \to \mathbb{R}$*

$$\hat{f} = \text{MLP} \circ \mathcal{L}_{\mathcal{G},\mathbf{W}}^{\text{conv}} \, ; \, \mathcal{L}_{\mathcal{G},\mathbf{W}}^{\text{conv}}(\mathbf{X}) \stackrel{\text{def.}}{=} \tilde{\mathbf{A}}\mathbf{X}\mathbf{W}, \tag{6}$$

*where $\mathbf{W} \in \mathbb{R}^{D \times D}$ is a matrix, $\text{MLP} \in \mathcal{MLP}_{N \times D}$ and $\tilde{\mathbf{A}}$ is the degree-normalized adjacency matrix for the input graph $\mathcal{G}$. The class $\mathcal{F}_{\mathcal{G},\mathbf{W}}$ is a stylized model class consisting of a graph convolution layer followed by an MLP, without a residual connection.*

We benchmark this class against a simplified version of our SMPNN block.

**Definition 4.4** (Residual Connection Model Class). *Again, fix $\mathbf{W} \in \mathbb{R}^{D \times D}$ and a graph $\mathcal{G}$ on $N$ nodes. This class, denoted by $\mathcal{F}_{\mathcal{G},\mathbf{W}}^{\text{residual}}$, consists of maps $\hat{f} : \mathbb{R}^{N \times D} \to \mathbb{R}$ with representation*

$$\hat{f} = \text{MLP} \circ \mathcal{L}_{\mathcal{G},\mathbf{W}}^{\text{conv+r}} \, ; \, \mathcal{L}_{\mathcal{G},\mathbf{W}}^{\text{conv+r}}(\mathbf{X}) \stackrel{\text{def.}}{=} \tilde{\mathbf{A}}\mathbf{X}\mathbf{W} + \mathbf{X}. \tag{7}$$

We emphasize that the MLPs within both sets, $\mathcal{F}_{\mathcal{G},\mathbf{W}}$ and $\mathcal{F}_{\mathcal{G},\mathbf{W}}^{\text{residual}}$, are identical. Hence, there are no inherent advantages or disadvantages between them; the sole discrepancy lies in the presence or absence of a residual connection.

The following set of results provides a counterexample to the universality of $\mathcal{F}_{\mathcal{G},\mathbf{W}}$ when $\mathcal{G}$ is a complete graph with self-loops (worst-case analysis) and further extends it to general graphs (like those found in practice). We show that this deficit is not an issue when incorporating a residual connection as $\mathcal{F}_{\mathcal{G},\mathbf{W}}^{\text{residual}}$ is universal.

**Theorem 4.5** (No Universal Approximation via Graph Convolution Alone). *Let $N, D$ be positive integers with $N \geq 2$. Let $\mathcal{G}$ be a complete graph (with self-loops) on $N$ nodes. For any weight matrix $\mathbf{W} \in \mathbb{R}^{D \times D}$, the class $\mathcal{F}_{\mathcal{G},\mathbf{W}}$ defined in equation 6 is not a universal approximator in $\mathcal{C}(\mathbb{R}^{N \times D})$.*

This is contrasted against our *positive result*, which shows that the universal approximation property of MLPs is preserved when passing input features through graph convolution with a residual connection.

**Lemma 4.6** (Injectivity criterion on the self-looped complete graph with residuals). *Let $N, D \geq 1$, let $\tilde{\mathbf{A}} = J_N/N$, and define $\mathcal{L}_{\mathcal{G},\mathbf{W}}^{\text{conv+r}}(\mathbf{X}) := \tilde{\mathbf{A}}\mathbf{X}\mathbf{W} + \mathbf{X}$. Then $\mathcal{L}_{\mathcal{G},\mathbf{W}}^{\text{conv+r}}$ is injective if and only if $I_D + \mathbf{W}$ is invertible, equivalently $-1 \notin \sigma(\mathbf{W})$.*

**Theorem 4.7** (Universality with residual connections). *Let $\zeta$ satisfy Assumption 4.2. Let $N, D \geq 1$, let $\tilde{\mathbf{A}} = J_N/N$, and let $\mathbf{W} \in \mathbb{R}^{D \times D}$. If $-1 \notin \sigma(\mathbf{W})$, then the class $\mathcal{F}_{\mathcal{G},\mathbf{W}}^{\text{residual}}$ is dense in $\mathcal{C}(\mathbb{R}^{N \times D})$ for the topology of uniform convergence on compact sets. In particular, if $\mathbf{W}$ has an absolutely continuous distribution on $\mathbb{R}^{D \times D}$ (for example, i.i.d. Gaussian entries), then this holds with probability one.*

The residual connections restore universality of the model class, while the pathological case where injectivity fails occurs with probability zero and is therefore (extremely) unlikely to arise in practice at initialization. Finally, we extend our results to general graphs like those found in experiments.

**Corollary 4.8** (Sufficient condition for universality on general graphs). *Let $\mathcal{G}$ be any graph with normalized adjacency matrix $\tilde{\mathbf{A}} \in \mathbb{R}^{N \times N}$, and let $\mathbf{W} \in \mathbb{R}^{D \times D}$. If the spectral norm*

$$\|\tilde{\mathbf{A}}\|_2 \, \|\mathbf{W}\|_2 < 1,$$

*then $\mathcal{L}_{\mathcal{G},\mathbf{W}}^{\text{conv+r}}$ is injective, and therefore the model class $\mathcal{F}_{\mathcal{G},\mathbf{W}}^{\text{residual}}$ is dense in $\mathcal{C}(\mathbb{R}^{N \times D})$.*

Thus, in the presence of residual connections, the expressivity of the architecture is linked to the scale of the weights $\mathbf{W}$ relative to the graph connectivity operator $\tilde{\mathbf{A}}$. This could inspire initialization techniques to guarantee that at the onset of training, the model resides in a "universal" regime (which depends on the input graph) where node-level information is preserved rather than collapsed.

## 5 EXPERIMENTAL VALIDATION

We experimentally validate our architecture and compare it to recent state-of-the-art baselines. We follow the evaluation protocol used in NodeFormer (Wu et al., 2022), DIFFormer (Wu et al., 2023a),

and SGFormer (Wu et al., 2023b). Additionally, we perform ablations and further experiments to verify the theoretical analysis presented in Section 4. For details, refer to Appendix E.

**Large Scale Graph Datasets** (Table 2). We compare the performance of SMPNNs to that of Graph Transformers tailored for large graph transductive learning, as well as other GNN baselines. Note that most Graph Transformer architectures (Table 1) are difficult to scale to these datasets. We observe that SMPNNs consistently outperform SOTA architectures without the need for attention. Augmenting the base SMPNN model with linear attention (Appendix B) leads to improvements in performance of under $1\%$ only, while substantially increasing computational overhead. For instance, in the case of the ogbn-products dataset, using the same hyperparameters, adding linear global attention with a single head increases the total number of model parameters from 834K to 2.4M. The resulting performance gain of only $0.18\%$ thus requires more than twice as many parameters.

Table 2: Test results (mean ± standard deviation over 5 runs) on large graph datasets. We report ROC-AUC for ogbn-proteins and accuracy for all others, higher is better.

| Dataset | ogbn-proteins | pokec | ogbn-arxiv | ogbn-products |
|---|---|---|---|---|
| Nodes | 132,534 | 1,632,803 | 169,343 | 2,449,029 |
| Edges | 39,561,252 | 30,622,564 | 1,166,243 | 61,859,140 |
| MLP | 72.04 ± 0.48 | 60.15 ± 0.03 | 55.50 ± 0.23 | 63.46 ± 0.10 |
| GCN (Kipf & Welling, 2017) | 72.51 ± 0.35 | 62.31 ± 1.13 | 71.74 ± 0.29 | 83.90 ± 0.10 |
| SGC (Wu et al., 2019) | 70.31 ± 0.23 | 52.03 ± 0.84 | 67.79 ± 0.27 | 81.21 ± 0.12 |
| GCN-NSampler Kipf & Welling (2017); Zeng et al. (2020) | 73.51 ± 1.31 | 63.75 ± 0.77 | 68.50 ± 0.23 | 83.84 ± 0.42 |
| GAT-NSampler Veličković et al. (2018); Zeng et al. (2020) | 74.63 ± 1.24 | 62.32 ± 0.65 | 67.63 ± 0.23 | 85.17 ± 0.32 |
| SIGN (Frasca et al., 2020) | 71.24 ± 0.46 | 68.01 ± 0.25 | 70.28 ± 0.25 | 80.98 ± 0.31 |
| NodeFormer (Wu et al., 2022) | 77.45 ± 1.15 | 68.32 ± 0.45 | 59.90 ± 0.42 | 87.85 ± 0.24 |
| DIFFormer (Wu et al., 2023a) | 79.49 ± 0.44 | 69.24 ± 0.76 | 64.94 ± 0.25 | 84.00 ± 0.07 |
| SGFormer (Wu et al., 2023b) | *79.53 ± 0.38* | *73.76 ± 0.24* | *72.63 ± 0.13* | *89.09 ± 0.10* |
| **SMPNN** | 83.15 ± 0.24 | 79.76 ± 0.19 | 73.75 ± 0.24 | 90.61 ± 0.05 |
| **SMPNN + Attention** | **83.65 ± 0.35** | **80.09 ± 0.12** | **74.38 ± 0.16** | **90.79 ± 0.04** |
| Linear Transformer | 60.87 ± 7.23 | 62.72 ± 0.09 | 58.02 ± 0.21 | 79.85 ± 0.08 |

Most Graph Transformers in the literature use multiple attention heads (Table 1); we experimented with up to 4 attention heads but found differences in terms of performance not to be statistically significant, which seems to align with the conclusions of (Wu et al., 2023b). Furthermore, note that as reported in the OGB benchmark paper (Hu et al., 2020), the Max Strongly Connected Component Ratio (MaxSCC Ratio) for the node-level datasets used in these experiments is 1.00 for all datasets (except for ogbn-products, which is 0.97). This indicates that the entire graph is a single strongly connected component, meaning every node is reachable from every other node. Although this does not guarantee that information bottlenecks will not be present, a high MaxSCC Ratio denotes high inter-connectivity, where information or influence can rapidly spread among a large proportion of the network, which may explain why attention is not that important in such settings. Apart from SMPNN and SMPNN augmented with attention, we also include a Linear Transformer baseline, which corresponds to removing the GCN layer from our architecture and substituting local convolution with global linear attention. This also performs worse than SMPNNs, which suggests that in these datasets, the locality inductive bias is important.

**100M Node Graph Dataset and Scalability Experiments** (Table 3). We test the scalability of our pipeline on the ogbn-papers-100M dataset. Our main competitors are SIGN (Frasca et al., 2020) and SGFormer (Wu et al., 2023b), as other Graph Transformers have not been able to scale to this dataset. We outperform SGFormer without requiring attention, again demonstrating scalability.

Table 3: Test results (mean accuracy ± standard deviation) on ogbn-papers-100M dataset.

| Dataset | ogbn-papers-100M |
|---|---|
| Nodes | 111,059,956 |
| Edges | 1,615,685,872 |
| MLP | 47.24 ± 0.31 |
| GCN (Kipf & Welling, 2017) | 63.29 ± 0.19 |
| SGC (Wu et al., 2019) | 63.29 ± 0.19 |
| GCN-NSampler Kipf & Welling (2017); Zeng et al. (2020) | 62.04 ± 0.27 |
| GAT-NSampler Veličković et al. (2018); Zeng et al. (2020) | 63.47 ± 0.39 |
| SIGN (Frasca et al., 2020) | *65.11 ± 0.14* |
| SGFormer (Wu et al., 2023b) | 66.01 ± 0.37 |
| **SMPNN** | **66.21± 0.10** |

**Additional Image, Text, and Spatio-Temporal Benchmarks** (Tables 4 and 5). For completeness, we also test our architecture on image and text classification with low label rates, as well as on spatio-temporal dynamics prediction, following (Wu et al., 2023a). Our model achieves the best

performance in most configurations for the CIFAR, STL, and 20News datasets. Likewise, in Table 5, SMPNNs are also competitive in spatio-temporal prediction. These results demonstrate that our architecture is applicable to a variety of tasks.

Table 4: Test results (mean accuracy ± standard deviation over 5 runs) on Image and Text datasets.

| Dataset | CIFAR | | | STL | | | 20News | | |
|---|---|---|---|---|---|---|---|---|---|
| Labels | 100 | 500 | 1000 | 100 | 500 | 1000 | 1000 | 2000 | 4000 |
| MLP | 65.9 ± 1.3 | 73.2 ± 0.4 | 75.4 ± 0.6 | 66.2 ± 1.4 | 73.0 ± 0.8 | 75.0 ± 0.8 | 54.1 ± 0.9 | 57.8 ± 0.9 | 62.4 ± 0.6 |
| ManiReg | 67.0 ± 1.9 | 72.6 ± 1.2 | 74.3 ± 0.4 | 66.5 ± 1.9 | 72.5 ± 0.5 | 74.2 ± 0.5 | 56.3 ± 1.2 | 60.0 ± 0.8 | 63.6 ± 0.7 |
| GCN-kNN | 66.7 ± 1.5 | 72.9 ± 0.4 | 74.7 ± 0.5 | 66.9 ± 0.5 | 72.1 ± 0.8 | 73.7 ± 0.4 | 56.1 ± 0.6 | 60.6 ± 1.3 | 64.3 ± 1.0 |
| GAT-kNN | 66.0 ± 2.1 | 72.4 ± 0.5 | 74.1 ± 0.5 | 66.5 ± 0.8 | 72.0 ± 0.8 | 73.9 ± 0.6 | 55.2 ± 0.8 | 59.1 ± 2.2 | 62.9 ± 0.7 |
| DenseGAT | OOM | OOM | OOM | OOM | OOM | OOM | 54.6 ± 0.2 | 59.3 ± 1.4 | 62.4 ± 1.0 |
| GLCN | 66.6 ± 1.4 | 72.8 ± 0.5 | 74.7 ± 0.3 | 66.4 ± 0.8 | 72.4 ± 1.3 | 74.3 ± 0.7 | 56.2 ± 0.8 | 60.2 ± 0.7 | 64.1 ± 0.8 |
| DIFFormer-a | 69.3 ± 1.4 | 74.0 ± 0.6 | 75.9 ± 0.3 | 66.8 ± 1.1 | 72.9 ± 0.7 | 75.3 ± 0.6 | 57.9 ± 0.7 | 61.3 ± 1.0 | 64.8 ± 1.0 |
| DIFFormer-s | 69.1 ± 1.1 | 74.8 ± 0.5 | 76.6 ± 0.3 | 67.8 ± 1.1 | 73.7 ± 0.6 | 76.4 ± 0.5 | 57.7 ± 0.3 | 61.2 ± 0.6 | 65.9 ± 0.8 |
| SMPNN | 68.6 ± 1.8 | 76.2 ± 0.5 | 78.0 ± 0.3 | 67.9 ± 0.9 | 73.9 ± 0.7 | 76.7 ± 0.5 | 58.9 ± 0.8 | 62.7 ± 0.6 | 65.6 ± 0.6 |

Table 5: Test results (mean MSE ± standard deviation over 5 runs) for spatio-temporal dynamics prediction datasets. Lower is better. *w/o g* stands for *without graph*.

| Dataset | Chickenpox | Covid | WikiMath |
|---|---|---|---|
| MLP | 0.924 ± 0.001 | 0.956 ± 0.198 | 1.073 ± 0.042 |
| GCN | 0.923 ± 0.001 | 1.080 ± 0.162 | 1.292 ± 0.125 |
| GAT | 0.924 ± 0.002 | 1.052 ± 0.336 | 1.339 ± 0.073 |
| GCN-kNN | 0.936 ± 0.004 | 1.475 ± 0.560 | 1.023 ± 0.058 |
| GAT-kNN | 0.926 ± 0.004 | 0.861 ± 0.123 | 0.882 ± 0.015 |
| DenseGAT | 0.935 ± 0.005 | 1.524 ± 0.319 | 0.826 ± 0.070 |
| DIFFormer-s | 0.914 ± 0.006 | 0.779 ± 0.037 | 0.731 ± 0.007 |
| DIFFormer-a | 0.915 ± 0.008 | 0.757 ± 0.048 | 0.763 ± 0.020 |
| DIFFormer-s w/o g | 0.916 ± 0.006 | 0.779 ± 0.028 | 0.727 ± 0.025 |
| DIFFormer-a w/o g | 0.916 ± 0.006 | 0.741 ± 0.052 | 0.716 ± 0.030 |
| SMPNN | 0.916 ± 0.006 | 0.756 ± 0.048 | 0.713 ± 0.032 |

**Ablation on SMPNN Architecture Components.** In Table 6, we conduct an ablation study. First, we ablate the standard SMPNN architecture by removing residual connections after graph convolution layers. In line with the theory presented in Section 4, we observe that this indeed has a significant impact on model performance. Next, we test the effect of fixing the learnable scaling coefficients ($\alpha_1^{(l)} = \alpha_2^{(l)} = 1, \forall l$) while retaining the residuals. This decreases performance for ogbn-proteins and ogbn-arxiv, whereas it slightly increases performance for pokec and ogbn-products. We decide to keep the scalings since they make the model more expressive in general. Additionally, we test removing the pointwise feedforward transformation, which leads to a slight drop in performance for all datasets and seems particularly relevant for ogbn-proteins. This experiment suggests that the most critical part of the architecture is the message-passing component, as expected. Lastly, for completeness, we also test removing the LayerNorm before the GCN, which generally leads to a drop in performance. However, in the case of ogbn-arxiv, this configuration obtains a test accuracy of $74.46\%$, which outperforms all other models and baselines. In conclusion, there is no free lunch; depending on the dataset, slight modifications may lead to improved performance and, interestingly, have even more of an effect than augmenting SMPNNs with linear attention. Nevertheless, we find that, in general, the standard SMPNN block is robust and leads to SOTA results across several datasets.

Table 6: Results (mean test set accuracy ± standard deviation over 5 runs) for ablation studies on OGBN and pokec large graph datasets.

| Model | Removed | ogbn-proteins | pokec | ogbn-arxiv | ogbn-products |
|---|---|---|---|---|---|
| SGFormer (Wu et al., 2023b) | N/A | 79.53 ± 0.38 | 73.76 ± 0.24 | 72.63 ± 0.13 | 89.09 ± 0.10 |
| SMPNN | N/A | 83.15 ± 0.24 | 79.76 ± 0.19 | 73.75 ± 0.24 | 90.61 ± 0.05 |
| SMPNN | Residual | 68.49 ± 2.59 | 68.17 ± 8.22 | 39.67 ± 14.60 | 89.69 ± 0.05 |
| SMPNN | $\alpha$ | 82.90 ± 0.34 | 80.10 ± 0.12 | 73.00 ± 0.59 | 90.77 ± 0.04 |
| SMPNN | FF | 80.51 ± 0.61 | 78.40 ± 0.20 | 73.25 ± 0.58 | 90.01 ± 0.10 |
| SMPNN | GCN LayerNorm | 80.74 ± 0.91 | 79.42 ± 0.15 | 74.46 ± 0.22 | 90.54 ± 0.08 |

**Deep Models are Possible with SMPNNs.** It is well known that conventional message-passing GNNs (Kipf & Welling, 2017; Veličković et al., 2018; Brody et al., 2021) are restricted to shallow architectures, since their performance degrades when stacking many layers. Next, we demonstrate that deep models are possible with SMPNNs. In particular, we perform an experiment in which we progressively increase the number of SMPNN layers while keeping all other hyperparameters constant for the ogbn-arxiv and ogbn-proteins datasets, from 2 to 12 layers (Table 7). Adding up to 6 layers seems to improve performance, and it plateaus thereafter. Additionally, we perform another experiment in which we follow the same procedure but for an SMPNN without residual connections after convolutions. In this case, we observe a clear drop in performance after 4 layers, which aligns with our theoretical understanding from Section 4.

Table 7: Results (mean accuracy ± standard deviation over 5 runs) on deep SMPNN architectures.

| Model | Removed | ogbn-arxiv, No. layers | | | | | | |
|-------|---------|----|----|----|----|----|----|----|
| | | 2 | 3 | 4 | 6 | 8 | 10 | 12 |
| **SMPNN** | N/A | 73.18 ± 0.30 | 73.33 ± 0.38 | 73.65 ± 0.49 | 73.75 ± 0.24 | 73.71 ± 0.54 | 73.68 ± 0.51 | 73.73 ± 0.50 |
| SMPNN | Residual | 72.56 ± 0.91 | 72.61 ± 0.83 | 71.38 ± 1.48 | 39.67 ± 14.60 | 25.87 ± 1.43 | 26.10 ± 1.68 | 26.59 ± 2.25 |
| Model | Removed | ogbn-proteins, No. layers | | | | | | |
| | | 2 | 3 | 4 | 6 | 8 | 10 | 12 |
| **SMPNN** | N/A | 81.44 ± 0.36 | 82.57 ± 0.45 | 82.64 ± 0.73 | 83.15 ± 0.24 | 83.36 ± 0.34 | 83.28 ± 0.36 | 83.45 ± 0.41 |
| SMPNN | Residual | 76.83 ± 1.56 | 68.93 ± 2.41 | 69.98 ± 3.08 | 68.49 ± 2.59 | 64.22 ± 1.22 | 65.54 ± 0.99 | 65.90 ± 1.35 |

**GPU Memory Scaling.** We proceed to examine the model's scalability utilizing the ogbn-products dataset and randomly sampling a subset of nodes for graph mini-batching (from 10K to 100K), following the analysis in (Wu et al., 2023b). In Figure 2 (Appendix F), we report the maximum GPU memory usage in GB against the number of edges in the subgraph induced by the sampled nodes. We can see that it asymptotes towards linearity in the number of edges as discussed in Section 3.2. For small subgraphs $N$ dominates the complexity term and results in a steeper slope. Additionally, we compare the computational overhead of different SMPNN configurations to that of SGFormer on a Tesla V100 GPU, see Figure 3 (Appendix F). Interestingly, the scaling factors $\alpha$ seem to have a substantial computational impact as the number of nodes increases, which, combined with the ablations in Table 6, suggests it may be better to remove them for very large graphs. Moreover, we find that the GPU consumption of SMPNNs can be substantially reduced when omitting the pointwise feedforward part of the block. This reduces computation cost substantially below that of SGFormer. Hence, one can trade GPU usage for performance: note that as shown in Table 6, SMPNNs without feedforwards still outperform the SGFormer baseline. SGFormer does not use feedforward networks or traditional attention mechanisms. Instead, it employs a simplified linear version of attention and even omits the softmax activation function.

## 6 CONCLUSION

Our SMPNN framework demonstrates that standard graph convolution, when packaged within a Pre-LN Transformer-style residual block, provides a simple yet effective architecture. We empirically show that SMPNNs enable deep message-passing networks without the degradation traditionally observed in GNNs and scale efficiently to large graphs, achieving strong empirical performance and outperforming recent scalable Graph Transformers without relying on global attention. We complement these results with a new universality-based theoretical analysis explaining why residualized convolution preserves expressivity. We do not claim to introduce a fundamentally new primitive.

Our results suggest that attention is often unnecessary (or provides only marginal gains) on "traditional" large transductive graphs. This may be partly explained by the local nature and high MaxSCC ratios of commonly used benchmarks. However, more broadly and importantly, the limited impact of attention in these settings may also arise from the absence of positional encodings, which are rarely used in large-graph models due to their computational cost (e.g., Laplacian positional encodings). Without positional signals to distinguish node positions, attention mechanisms may effectively degenerate into computing near-average feature aggregations. This consideration is likely important for emerging long-range graph benchmarks (Liang et al., 2026; Bamberger et al., 2025), which we leave for future work.

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

## A  ADDITIONAL BACKGROUND ON GRAPH REPRESENTATION LEARNING

**Graph Representation Learning and Notation.** We denote a graph with $\mathcal{G} = (\mathcal{V}, \mathcal{E})$, where $\mathcal{V}$ represents a set of nodes (or vertices), and $\mathcal{E} \subseteq (\mathcal{V} \times \mathcal{V})$ is a set of 2-tuples signifying edges (or links) within the graph. For any pair of nodes $v_i$ and $v_j$ within $\mathcal{G}$, their connection is encoded as $(v_i, v_j) \in \mathcal{E}$ if the edge originates from $v_i$ and terminates at $v_j$. The (one-hop) neighborhood of node $v_i$ constitutes the set of nodes sharing an edge with $v_i$, denoted by $\mathcal{N}(v_i) = \{v_j | (v_i, v_j) \in \mathcal{E}\}$. To encode the structural connectivity amongst nodes in a graph of $N = |\mathcal{V}|$ nodes and $E = |\mathcal{E}|$ edges, an adjacency matrix $\mathbf{A} \in \mathbb{R}^{N \times N}$ is employed. This adjacency matrix may adopt a weighted or unweighted representation. In the case of a weighted adjacency matrix, the entries $A_{ij} \in \mathbb{R}$ symbolize the strength of the connection, with $A_{ij} = 0$ denoting absence of a connection if $(v_i, v_j) \notin \mathcal{E}$. Conversely, in an unweighted adjacency matrix, $A_{ij} = 1$ signifies the presence of an edge, and $A_{ij} = 0$ otherwise. The degree matrix $\mathbf{D} \in \mathbb{R}^{N \times N}$ is defined as the matrix where each entry on the diagonal is the row-sum of the adjacency matrix: $D_{ii} = \sum_j A_{ij}$. Based on this, we can define the graph Laplacian as $\mathbf{L} = \mathbf{D} - \mathbf{A}$. The normalized graph Laplacian, denoted as $\mathbf{L}_{\text{norm}} = \mathbf{I} - \mathbf{D}^{-\frac{1}{2}} \mathbf{A} \mathbf{D}^{-\frac{1}{2}}$, is a variation of the graph Laplacian that takes into account the degrees of the nodes. Additionally, within our context, we are interested in graphs with node features. Each node $v_i \in \mathcal{V}$ is accompanied by a $D$-dimensional feature vector $\mathbf{x}_i \in \mathbb{R}^{1 \times D}$. The feature vectors for all nodes within the graph can be aggregated into a single matrix $\mathbf{X} \in \mathbb{R}^{N \times D}$ by stacking them along the row dimension.

**Message Passing Neural Networks (MPNNs)** are a class of Graph Neural Networks (GNNs). MPNNs operate by means of message passing, wherein nodes exchange vector-based messages to refine their representations, leveraging the graph connectivity structure as a geometric prior. Following Bronstein et al. (2021), a message passing GNN layer $l$ (excluding edge and graph-level features for simplicity) over a graph $\mathcal{G}$ is defined as: $\mathbf{x}_i^{(l+1)} = \phi\left(\mathbf{x}_i^{(l)}, \bigoplus_{j \in \mathcal{N}(v_i)} \psi(\mathbf{x}_i^{(l)}, \mathbf{x}_j^{(l)})\right)$, where $\psi$ denotes a message passing function, $\bigoplus$ is some permutation-invariant aggregation operator, and $\phi$ is a readout or update function. Both $\psi$ and $\phi$ are learnable and typically implemented as MLPs. It is important to note that the update equation is *local*, and that at each layer each node only communicates with its one-hop neighborhood given by the adjacency matrix of the graph. More distant nodes are accessed by stacking multiple message-passing layers.

## B  AUGMENTING SMPNNS WITH ATTENTION

SMPNNs, in principle, perform local convolution but can also be enhanced with attention. Here, we discuss the specific attention mechanism implemented in our experiments.

**Linear Global Attention.** Global attention can be seen as computing message passing over a fully connected graph. In our block, attention is computed with respect to a virtual node rather than through full $\mathcal{O}(N^2)$ pairwise attention, which is more suitable for scaling to large graphs. Given the input query, key, and value matrices,

$$\mathbf{Q} = \mathbf{X}^{(l)} \mathbf{W}_Q \in \mathbb{R}^{N \times D}, \qquad \mathbf{K} = \mathbf{X}^{(l)} \mathbf{W}_K \in \mathbb{R}^{N \times D}, \qquad \mathbf{V} = \mathbf{X}^{(l)} \mathbf{W}_V \in \mathbb{R}^{N \times D},$$

we apply the following operations. First, we compute a summed query vector incorporating contributions from all nodes, and normalize both this summed query vector and the keys:

$$\mathbf{Q}_{\text{sn}} = \frac{\sum_{i=1}^{N} \mathbf{Q}_i}{\left\|\sum_{i=1}^{N} \mathbf{Q}_i\right\|_2} \in \mathbb{R}^{1 \times D}, \qquad \mathbf{K}_{n,i} = \frac{\mathbf{K}_i}{\|\mathbf{K}_i\|_2} \in \mathbb{R}^{1 \times D}, \tag{8}$$

for each node $i = 1, \ldots, N$. Based on these, we compute attention weights

$$\mathbf{A}_{\text{attention}} = \text{softmax}\left(\mathbf{Q}_{\text{sn}} \mathbf{K}_n^T\right) \in \mathbb{R}^{1 \times N}, \tag{9}$$

where $\mathbf{K}_n \in \mathbb{R}^{N \times D}$ is the matrix whose $i$-th row is $\mathbf{K}_{n,i}$. We then use these weights to aggregate the values into a single global feature vector

$$\mathbf{g}^{(l)} = \mathbf{A}_{\text{attention}} \mathbf{V} \in \mathbb{R}^{1 \times D}, \tag{10}$$

which is broadcast to all nodes:

$$\mathbf{X}^{(l+1)} = \mathbf{1}_N \mathbf{g}^{(l)} \in \mathbb{R}^{N \times D}, \tag{11}$$

where $\mathbf{1}_N \in \mathbb{R}^{N \times 1}$ is a column vector of ones. Equivalently,

$$\mathbf{X}^{(l+1)} = (\mathbf{1}_N \mathbf{A}_{\text{attention}}) \mathbf{V}, \tag{12}$$

where $\mathbf{1}_N \mathbf{A}_{\text{attention}} \in \mathbb{R}^{N \times N}$ is a matrix with all rows equal to the same attention vector.

The operations above extend naturally to multi-head attention. If we use $H$ attention heads, then for each head $h$ we compute

$$\mathbf{A}^{(h)}_{\text{attention}} = \text{softmax}\left( \mathbf{Q}^{(h)}_{\text{sn}} (\mathbf{K}^{(h)}_n)^T \right) \in \mathbb{R}^{1 \times N}, \tag{13}$$

and the corresponding head output

$$\mathbf{g}^{(h)} = \mathbf{A}^{(h)}_{\text{attention}} \mathbf{V}^{(h)} \in \mathbb{R}^{1 \times D}. \tag{14}$$

The final global feature matrix is then obtained by aggregating (using the mean) the contributions from all heads:

$$\mathbf{X}^{(l+1)} = \mathbf{1}_N \left( \frac{1}{H} \sum_{h=1}^{H} \mathbf{g}^{(h)} \right) \in \mathbb{R}^{N \times D}. \tag{15}$$

**Message-Passing with Parallel Attention.** To augment SMPNN we use the following procedure:

$$\mathbf{H}^{(l)}_{1,local} = \text{LayerNorm}(\mathbf{X}^{(l)}). \tag{16}$$

$$\mathbf{H}^{(l)}_{2,local} = \text{SiLU}\left( \tilde{\mathbf{A}} \mathbf{H}^{(l)}_{1,local} \mathbf{W}^{(l)}_1 \right), \tag{17}$$

$$\mathbf{H}^{(l)}_{1,global} = \text{LayerNorm}(\mathbf{X}^{(l)}), \tag{18}$$

$$\mathbf{H}^{(l)}_{2,global} = \text{LinearGlobalAttention}\left( \mathbf{H}^{(l)}_{1,global} \right), \tag{19}$$

$$\mathbf{H}^{(l)}_2 = \alpha^{(l)}_1 \left( \mathbf{H}^{(l)}_{2,local} + \mathbf{H}^{(l)}_{2,global} \right) + \mathbf{X}^{(l)}, \tag{20}$$

where LayerNorms have different parameters for local and global features. Pointwise-feedforward operations are kept as in the original SMPNN formulation.

The update equations above are reminiscent of hybrid Graph Transformers (Rampášek et al., 2022; Wu et al., 2022; 2023a;b), and we find that they do not provide tangible improvements over standard SMPNNs. Note that we also experimented with other attention mechanisms, which did not help.

## C  Oversmoothing and Residual Connections

Oversmoothing is regarded as the tendency of node features to approach the same value after several message-passing transformations. This phenomenon has prompted the adoption of relatively shallow GNNs, as adding many layers has traditionally resulted in node-level features that are too similar to each other and, hence, indistinguishable for downstream learners. This has hindered scalability and led to models with orders of magnitude fewer parameters than, for instance, counterparts in the literature on 2D generative modeling (Esser et al., 2024) and LLMs (Touvron et al., 2023), which have already adopted architectures with billions of parameters.

In general, standalone message-passing GNNs tend to behave as low-pass filters and may consequently lead to oversmoothing. However, this does not always need to be the case: graph convolution can also magnify high frequencies if it is augmented with residual connections. Following (Giovanni et al., 2023), message-passing is characterized as being *low-frequency dominant* (LFD) or *high-frequency dominant* (HFD) depending on the asymptotic behavior of the normalized Dirichlet energy of the system as the number of diffusion layers $l \to \infty$. These lead to oversmoothing and oversharpening, respectively.

**Definition C.1** (Graph Dirichlet Energy of a Message Passing System (Zhou & Scholkopf, 2005)).
*We define the Dirichlet Energy as a map* $\mathfrak{E}^{Dir} : \mathbb{R}^{N \times D} \to \mathbb{R}$

$$\mathfrak{E}^{Dir}(\mathbf{X}) \overset{\text{def.}}{=} \frac{1}{2} \sum_{(i,j) \in \mathcal{E}} ||(\nabla \mathbf{X})_{ij}||^2 \, ; \, (\nabla \mathbf{X})_{ij} \overset{\text{def.}}{=} \frac{\mathbf{x}_j}{\sqrt{\deg(v_j)}} - \frac{\mathbf{x}_i}{\sqrt{\deg(v_i)}}, \tag{21}$$

*where $(\nabla \mathbf{X})_{ij}$ is the edge-wise gradient of the graph node features $\mathbf{X} \in \mathbb{R}^{N \times D}$.*

**Definition C.2** (Graph Frequency Dominance (Giovanni et al., 2023)). *We say a message passing GNN is LFD if its normalized Dirichlet energy $\mathfrak{E}^{Dir}(\mathbf{X}^{(l)})/||\mathbf{X}^{(l)}||^2 \to \lambda_1 = 0$ for $l \to \infty$ and HFD if $\mathfrak{E}^{Dir}(\mathbf{X}^{(l)})/||\mathbf{X}^{(l)}||^2 \to \lambda_N$ for $l \to \infty$, where $\mathbf{X}^{(l)}$ encapsulates the node-level features at a given layer $l$ and $1 \leq \lambda_N < 2$ (Chung, 1996) is the largest eigenvalue whose corresponding eigenvector of the normalized graph Laplacian captures fine-grained microscopic behavior of the signal on the graph.*

In particular, the above characterization relies on analyzing message-passing from the perspective of gradient flows in the limit of infinitely many layers. Although analyzing the asymptotic behavior of GNNs is interesting from a theoretical perspective, the normalized Dirichlet energy is not directly predictive of model success, since both over-smoothing and over-sharpening will lead to degradation in performance (Giovanni et al., 2023). We provide an alternative perspective through the lens of universal approximation, which does not rely on asymptotic behavior. Specifically, we demonstrate that the universal approximation property of downstream classifiers is compromised when passing input features through graph convolution layers, but it can be restored with a residual connection.

Moreover, by the results of Riedi et al. (2023), residual connections also make the loss landscape of deep learning models less jagged, which may aid optimization (which we do not tackle in this work).

## D  PROOFS

*Proof of Theorem 4.5.* For any $\mathbf{X} \in \mathbb{R}^{N \times D}$, define its row-average $\bar{\mathbf{x}} := \frac{1}{N} \mathbf{1}_N^\top \mathbf{X} \in \mathbb{R}^{1 \times D}$. Since $\tilde{\mathbf{A}} = J_N/N = (1/N)\mathbf{1}_N \mathbf{1}_N^\top$, we have $\tilde{\mathbf{A}}\mathbf{X} = \frac{1}{N}\mathbf{1}_N\mathbf{1}_N^\top \mathbf{X} = \mathbf{1}_N \bar{\mathbf{x}}$. Hence every row of $\tilde{\mathbf{A}}\mathbf{X}\mathbf{W}$ is identical. Therefore the image of $\mathcal{L}_{\mathcal{G},\mathbf{W}}^{\text{conv}}$ is contained in the proper subspace $\{\mathbf{1}_N \mathbf{v}^\top : \mathbf{v} \in \mathbb{R}^D\} \subsetneq \mathbb{R}^{N \times D}$, so $\mathcal{L}_{\mathcal{G},\mathbf{W}}^{\text{conv}}$ is not injective. Concretely, any two inputs with the same row-average produce the same output.

By the universal approximation theorem, the class $\mathcal{MLP}_{N \times D}^\zeta$ is dense in $\mathcal{C}(\mathbb{R}^{N \times D})$ for the topology of uniform convergence on compact sets. The map $\mathcal{L}_{\mathcal{G},\mathbf{W}}^{\text{conv}}$ is continuous but not injective. Therefore, by (Kratsios & Bilokopytov, 2020, Theorem 3.4), precomposition with $\mathcal{L}_{\mathcal{G},\mathbf{W}}^{\text{conv}}$ does not preserve density. $\qquad\square$

*Proof of Lemma 4.6.* Every $\mathbf{X} \in \mathbb{R}^{N \times D}$ admits a unique decomposition $\mathbf{X} = \mathbf{1}_N \mathbf{u}^\top + \mathbf{Y}$ with $\mathbf{1}_N^\top \mathbf{Y} = 0$ for some $\mathbf{u} \in \mathbb{R}^D$. Indeed, define the row-average vector $\mathbf{u}^\top = \frac{1}{N}\mathbf{1}_N^\top \mathbf{X}$ and set $\mathbf{Y} = \mathbf{X} - \mathbf{1}_N \mathbf{u}^\top$. Then $\mathbf{1}_N^\top \mathbf{Y} = 0$ by construction, and the decomposition is unique because matrices of the form $\mathbf{1}_N \mathbf{v}^\top$ (which have identical rows) intersect the subspace $\{\mathbf{Y} : \mathbf{1}_N^\top \mathbf{Y} = 0\}$ only at the zero matrix. Intuitively, this separates the node features into a global average signal $\mathbf{1}_N \mathbf{u}^\top$ and variations around the average $\mathbf{Y}$. Since $\tilde{\mathbf{A}} = J_N/N$, multiplication by $\tilde{\mathbf{A}}$ averages the rows of a matrix, yielding $\tilde{\mathbf{A}}(\mathbf{1}_N \mathbf{u}^\top) = \mathbf{1}_N \mathbf{u}^\top$ and $\tilde{\mathbf{A}}\mathbf{Y} = 0$. Applying the residual convolution operator $\mathcal{L}_{\mathcal{G},\mathbf{W}}^{\text{conv+r}}(\mathbf{X}) = \tilde{\mathbf{A}}\mathbf{X}\mathbf{W} + \mathbf{X}$ to the decomposition therefore gives $\mathcal{L}_{\mathcal{G},\mathbf{W}}^{\text{conv+r}}(\mathbf{X}) = \mathbf{1}_N \mathbf{u}^\top \mathbf{W} + \mathbf{1}_N \mathbf{u}^\top + \mathbf{Y} = \mathbf{1}_N \mathbf{u}^\top (I_D + \mathbf{W}) + \mathbf{Y}$. To determine injectivity it suffices to examine the kernel of this linear map. Setting $\mathcal{L}_{\mathcal{G},\mathbf{W}}^{\text{conv+r}}(\mathbf{X}) = 0$ yields $\mathbf{1}_N \mathbf{u}^\top (I_D + \mathbf{W}) + \mathbf{Y} = 0$. The first term has identical rows while $\mathbf{Y}$ has rows summing to zero, so they belong to complementary subspaces and must vanish separately. Hence $\mathbf{Y} = 0$ and $\mathbf{u}^\top (I_D + \mathbf{W}) = 0$. This system has only the trivial solution $\mathbf{u} = 0$ if and only if $I_D + \mathbf{W}$ is invertible. Since the eigenvalues of $I_D + \mathbf{W}$ are $1 + \lambda_i$ where $\lambda_i \in \sigma(\mathbf{W})$, this occurs precisely when $-1 \notin \sigma(\mathbf{W})$. $\qquad\square$

*Proof of Theorem 4.7.* By the universal approximation theorem, $\mathcal{MLP}_{N \times D}^{\zeta}$ is dense in $\mathcal{C}(\mathbb{R}^{N \times D})$ for the topology of uniform convergence on compact sets. By Lemma 4.6, $\mathcal{L}_{\mathcal{G},\mathbf{W}}^{\text{conv+r}}$ is continuous and injective whenever $-1 \notin \sigma(\mathbf{W})$. Hence (Kratsios & Bilokopytov, 2020, Theorem 3.4) implies that precomposition with $\mathcal{L}_{\mathcal{G},\mathbf{W}}^{\text{conv+r}}$ preserves density.

For the probabilistic statement, the exceptional set $\{\mathbf{W} : \det(I_D + \mathbf{W}) = 0\}$ is the zero set of a nonzero polynomial in the entries of $\mathbf{W}$, so it has Lebesgue measure zero. Therefore it is hit with probability zero by any absolutely continuous distribution. $\square$

*Proof of Corollary 4.8.* Let $\text{vec}(\cdot)$ denote the vectorization operator, which stacks the columns of a matrix $\mathbf{X} \in \mathbb{R}^{N \times D}$ into a vector in $\mathbb{R}^{ND}$. The vectorization identity $\text{vec}(\mathbf{AXB}) = (\mathbf{B}^\top \otimes \mathbf{A}) \text{vec}(\mathbf{X})$ gives $\text{vec}(\tilde{\mathbf{A}}\mathbf{XW}+\mathbf{X}) = (\mathbf{W}^\top \otimes \tilde{\mathbf{A}}) \text{vec}(\mathbf{X}) + \text{vec}(\mathbf{X}) = \left( I_{ND} + \mathbf{W}^\top \otimes \tilde{\mathbf{A}} \right) \text{vec}(\mathbf{X})$, where $\otimes$ is the Kronecker product. Thus injectivity of $\mathcal{L}_{\mathcal{G},\mathbf{W}}^{\text{conv+r}}$ is equivalent to invertibility of $I_{ND} + \mathbf{W}^\top \otimes \tilde{\mathbf{A}}$. If $\|\tilde{\mathbf{A}}\|_2 \|\mathbf{W}\|_2 < 1$, then $\|\mathbf{W}^\top \otimes \tilde{\mathbf{A}}\|_2 = \|\mathbf{W}^\top\|_2 \|\tilde{\mathbf{A}}\|_2 = \|\mathbf{W}\|_2 \|\tilde{\mathbf{A}}\|_2 < 1$, so $I_{ND} + \mathbf{W}^\top \otimes \tilde{\mathbf{A}}$ is invertible by the Neumann series. Hence the feature map is injective. Since the map is linear and therefore continuous, Theorem 3.4 of Kratsios & Bilokopytov (2020) implies that the class $\mathcal{F}_{\mathcal{G},\mathbf{W}}^{\text{residual}}$ is dense in $\mathcal{C}(\mathbb{R}^{N \times D})$. $\square$

# E   TRAINING DETAILS

**Dataset information.** All datasets used in this work are publicly available. For OGB datasets, see (Hu et al., 2020); for pokec, see (Leskovec & Krevl, 2014); and for the spatio-temporal datasets, see PyTorch Geometric Temporal (Rozemberczki et al., 2021). Following DIFFormer (Wu et al., 2023a), we perform experiments on both image and text datasets to test the applicability of SMPNNs to a variety of tasks. The procedure described next is exactly the same as in (Wu et al., 2023a). We use STL(-10) and CIFAR(-10) as our image datasets. For the first, we use all 13,000 images, each classified into one of 10 categories. In the case of the latter, CIFAR, the dataset was originally pre-processed in previous work using 1,500 images from each of the 10 categories, resulting in a total of 15,000 images. For both datasets, 10, 50, or 100 instances per class were randomly selected for the training set (we use the same random splits as in the baselines for reproducibility and to obtain a fair comparison), 1,000 instances for validation, and the remaining instances for testing.

**Train, validation, and test splits.** We use all public data splits when available, such as in the case of the OGB benchmarks (Hu et al., 2020). Otherwise we follow the data split in the literature (Wu et al., 2022; 2023a;b) for a consistent comparison. Note that the code for these baselines is available online, including torch seeds for datasets with random splits. We use the *exact same* seeds (`123` in most cases).

**Graph Sampling and Batching for Training and Inference.** For ogbn-arxiv, CIFAR, STL, 20News, Chickenpox, Covid, and WikiMath, we employ full-graph training. Specifically, we input the entire graph into the model and simultaneously predict all node labels for the loss computation using a single GPU. During inference, we also use the entire graph as input and calculate the evaluation metric based on the predictions for the validation and test set nodes. For the rest of the datasets (except ogbn-papers-100M discussed later), due to their large sizes, we adopt the mini-batch training approach of (Wu et al., 2022; 2023a;b). More concretely, the subsampling procedure of the Graph Transformer baselines uses the following approach for training: a batch of nodes from the large graph dataset is selected at random, and based on this, the induced subgraph is sampled by extracting the edges connecting the selected nodes from the full graph adjacency matrix (the induced subgraph is obtained using `subgraph` from `torch geometric utils`). For inference, the entire graph is fed into the model on CPU. For the exceptionally large ogbn-papers-100M graph, which cannot fit into CPU memory, we apply the same mini-batch partition strategy used during training for inference using a neighbor sampler with 3-hops and 15, 10, and 5 neighbors per hop using the standard `NeighborLoader` from `torch geometric`, in line with (Wu et al., 2023b).

**GPU Memory Scaling Experiments.** In these experiments, we use a single Tesla V100 GPU. We use the same training configuration for all networks: node batch sizes of 10K, 20K, 40K, 60K, 80K, and 100K (using `subgraph`), no weight decay, Adam optimizer with a learning rate of $10^{-3}$, and

dropout of $0.1$. We set all model configurations to have 256 hidden channels and 6 layers, modifying only the parts of the configuration discussed in the main text. For SGFormer, we increase the number of GNN layers to 6 and fix the linear attention layer to 1, as in the original paper (Wu et al., 2023b).

**Model Configurations.** For most of the large-scale graph experiments (unless otherwise stated), we use SMPNNs with 6 layers. Note that we test other configurations; for instance, in Table 7. For the image, text, and spatio-temporal datasets, we use 2 layers to avoid overfitting. The exact configurations can be found in the supplementary material.

# F    GPU MEMORY SCALING PLOTS

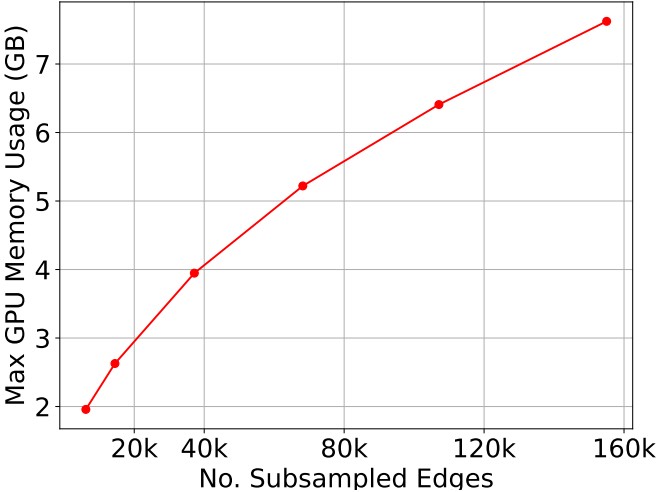

Figure 2: Max GPU consumption versus number of edges in the subgraph for SMPNN with 6 layers.

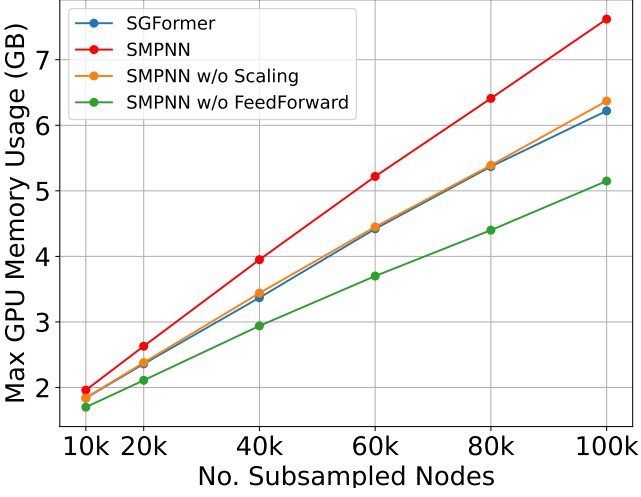

Figure 3: Max GPU consumption versus the number of nodes in the batch subgraph for different models with 6 layers.

