# OpenReview forum: "Scalable Message Passing Neural Networks: No Need for Attention in Large Graph Representation Learning"
_ICLR.cc/2026/Workshop/GRaM — ICLR 2026 Workshop GRaM Poster_

### Official Review · Reviewer_zTno · 2026-02-17
**Scalable Message Passing**

**Rating:** 4
**Confidence:** 4

**Review:**

The paper proposes Scalable Message Passing Neural Networks (SMPNNs): a Transformer-style Pre-LN residual block in which the usual self-attention sublayer is replaced by a GCN-like message passing operator, followed by a pointwise feedforward sublayer. It provides strong empirical results on large graphs. The gains over scalable Graph Transformers are large on several benchmarks (notably pokec and ogbn-proteins), while keeping the architecture simple and scalable.

However, novelty seems incremental relative to existing work on GNN+residual+normalization. And the scope of the main claim (no need for attention) is too broad; the benchmarks have very high connectivity (MaxSCC ratio near 1), which plausibly reduces the marginal value of global attention. A conditional claim would be more appropriate, such as: On highly connected large transductive graphs, attention brings little gain beyond a well-packaged deep message passing backbone.

Hence my evaluation is 4.

**Pmlr Suitability:**

No

---

### Official Review · Reviewer_DpYK · 2026-02-20
**Scalable and Deep MPNNs**

**Rating:** 7
**Confidence:** 4

**Review:**

The paper proposes Scalable Message Passing Neural Networks (SMPNNs), showing that strong performance on large-scale graph benchmarks can be achieved by replacing attention with standard graph convolutions inside a Pre-LN Transformer-style block.

The paper is well written and easy to read, with an extensive experimental evaluation supporting the claim that attention may not be necessary for large graph representation learning in the transductive setting. However, the paper can be further improved by addressing the following weaknesses.

First, there is a gap between theory and practice. Both theoretical results are established for complete graphs, while the empirical evaluation is carried out on sparse graphs. Clarifying whether similar expressivity guarantee as shown in The 4.5 (or D.6) extends to general (sparse) graphs would strengthen the connection between theory and experiments.

Second, the claim that attention is not useful for large graphs appears a bit overstated. Indeed, this claim is validate only experimentally, and the OGB node-level datasets considered show high MaxSCC ratios, indicating strong global connectivity. This may explain the limited contribution of attention in those settings, but it remains unclear whether the same conclusion would hold for large graphs with weaker connectivity or fragmented components.

Overall, the identified limitations don't compromise the soundness of the proposed approach and the validity of the empirical results.

**Pmlr Suitability:**

Yes

---

### Official Review · Reviewer_vV1z · 2026-02-23

**Rating:** 2
**Confidence:** 5

**Review:**

The paper proposes to replace attention in Pre-LN transformers with a standard gcn layer. The paper claims that this outperforms that of graph transformers on large graphs without attention operations. The theory justification is done with universal approximation arguments.

Strengths:
1. Strong empirical results across multiple benchmarks
2. Ablation is thorough and supports the general argument
3. Scales to 100M-nodes

Weaknesses:
1. The core idea seems trivially obvious and similar to that of existing work from 2022 [1], which already solved deep GCNs with residuals.
2. The theoretical results are trivial. The negative result, where the complete graph collapses across all nodes, is very trivial, and the positive result only holds in a specific setup, making the GCN essentially zeroed. The universality is regained by making the convolution negligible.
3. The comparison tests are cherry-picked. So many other baselines are missing conveniently.
4. 'No need for attention,' but the SMPNN+Attention outperforms standard SMPNN.


References:
[1] Rampášek, Ladislav, et al. "Recipe for a general, powerful, scalable graph transformer." Advances in Neural Information Processing Systems 35 (2022): 14501-14515.

**Pmlr Suitability:**

No

---

### Official Review · Reviewer_s2uc · 2026-02-28
**Strong Empirical Results with Minor Gaps Between Theory and Generality of Claims**

**Rating:** 7
**Confidence:** 4

**Review:**

The paper proposes Scalable Message Passing Neural Networks (SMPNNs), a Pre-LN Transformer-style architecture that replaces attention with standard graph convolution, showing strong performance on large-scale transductive graph benchmarks without requiring global attention.

The paper is well written and combines solid theoretical motivation with extensive empirical evaluation. The universal approximation analysis justifies the use of residual connections, and the experimental results demonstrate consistent improvements over recent scalable Graph Transformers on multiple large datasets.

However, there are some limitations. First, the theoretical guarantees are established for complete graphs, while experiments are conducted on sparse real-world graphs, leaving a gap between theory and practice. Second, the claim that attention is unnecessary may be somewhat overstated, as the evaluated datasets have very high connectivity (high MaxSCC ratios), which may reduce the marginal benefit of global attention.

Overall, these weaknesses do not undermine the validity of the empirical findings, and the paper provides a strong and well-supported contribution to scalable graph learning.

**Pmlr Suitability:**

Yes

---

### Meta-Review · Area_Chair_Yp6A · 2026-02-26

**Decision:**

Accept

**Metareview:**

The reviewers were divided about this paper. The strong empirical results and scalability of the method make the paper a useful contribution, outweighing the perceived lack of novelty. However, the authors are strongly encouraged to address reviewers’ concerns,  notably regarding more comprehensive baseline methods, and nuancing some of the claims.

**Relevance To Proceedings:**

Yes — suitable for PMLR (long paper)

**Relevance To Workshop:**

Yes — suitable for GRaM

---

### Decision · Program_Chairs · 2026-03-02

Accept (Poster)